# Inversion Method for Chlorophyll-a Concentration in High-Salinity Water Based on Hyperspectral Remote Sensing Data

**DOI:** 10.3390/s24134181

**Published:** 2024-06-27

**Authors:** Nan Wang, Zhiguo Wang, Pingping Huang, Yongguang Zhai, Xiangli Yang, Jianyu Su

**Affiliations:** 1College of Information Engineering, Inner Mongolia University of Technology, Hohhot 010080, China; 20211800117@imut.edu.cn (N.W.);; 2Inner Mongolia Key Laboratory of Radar Technology and Application, Hohhot 010051, China; 3College of Information Science and Engineering, Chongqing Jiaotong University, Chongqing 400074, China

**Keywords:** hyperspectral remote sensing data, Daihai water body, chlorophyll-a concentration, salinity

## Abstract

As one of the important lakes in the “One Lake and Two Seas” of the Inner Mongolia Autonomous Region, the monitoring of water quality in Lake Daihai has attracted increasing attention, and the concentration of chlorophyll-a directly affects the water quality, making the monitoring of chlorophyll-a concentration in Lake Daihai particularly crucial. Traditional methods of monitoring chlorophyll-a concentration are not only inefficient but also require significant human and material resources. Remote sensing technology has the advantages of wide coverage and short update cycles. For lakes such as Daihai with a high salinity content, salinity is considered a key factor when inverting the concentration of chlorophyll-a. In this study, machine learning models, including model stacking from ensemble learning, a ridge regression model, and a random forest model, were constructed. After comparing the training accuracy of the three models on Zhuhai-1 satellite data, the random forest model, which had the highest accuracy, was selected as the final training model. By comparing the accuracy changes before and after adding salinity factors to the random forest model, a high-precision model for inverting chlorophyll-a concentration in hypersaline lakes was obtained. The research results show that, without considering the salinity factor, the root mean square error (RMSE) of the model was 0.056, and the coefficient of determination (R^2^) was 0.64, indicating moderate model performance. After adding the salinity factor, the model accuracy significantly improved: the RMSE decreased to 0.047, and the R^2^ increased to 0.92. This study provides a solid basis for the application of remote sensing technology in hypersaline aquatic environments, confirming the importance of considering salinity when estimating chlorophyll-a concentration in hypersaline waters. This research helps us gain a deeper understanding of the water quality and ecosystem evolution in Daihai Lake.

## 1. Introduction

Chlorophyll-a is the most abundant pigment in plankton or algae, and its concentration in water can represent the biomass of the photosynthetic autotrophs of plankton and its primary productivity. Its content can directly reflect the eutrophication degree of water bodies, which is an important indicator for evaluating water quality conditions [1]. Chlorophyll, as an important component of algal cells, can be used to establish a connection between remote sensing images and chlorophyll-a concentration values, and it can be used to realize non-contact remote sensing estimation, which is helpful in analyzing and gradually solving the eutrophication problems in lakes [2]. In the inversion study of Chl-a concentration, a change in chlorophyll concentration will affect the reflectance of the input band [3]. Remote sensing technology can carry out large-scale water quality monitoring activities and discover the migration characteristics of pollution sources and pollutants that are difficult to reveal using conventional methods. It has outstanding advantages, for example, it is dynamic, has a large scale, and is fast. Therefore, it plays an increasingly important role in inland water quality monitoring [4].

In recent years, many scholars have conducted relevant research on the inversion of chlorophyll-a concentration in lakes based on different data sources. For instance, Xiyong Zhao et al. [5] used a multispectral camera mounted on a drone to collect data, and they built models using four machine learning algorithms to invert the chlorophyll-a concentration in Erhai Lake. The results showed that using vegetation indices for the multiple regression estimation of Chl-a concentration was generally superior to using single vegetation indices and original band information for regression. Elias S. Leggesse et al. [6] predicted chlorophyll-a (Chl-a) concentrations in freshwater bodies using six machine learning (ML) algorithms integrated with Landsat 8 images. Raphael M. Kudela et al. [7] used the Ocean and Land Color Instrument (OLCI) on Sentinel-3 to conduct a real-time image analysis of the current status of chlorophyll in San Francisco Bay, USA. Ali Reza Shahvaran et al. [8] paired 236 satellite scenes obtained from Landsat 5, 7, and 8 and Sentinel-2 between 2000 and 2022 with 600 nearly simultaneous and collocated in situ measurements of Chl-a concentration, and applied them to the western basin of Lake Ontario. The results showed that Sentinel-2 and Landsat 8 data provided the best results, while Landsat 5 and 7 data performed poorly. However, Yu Sheng et al. [9] used Sentinel-2 remote sensing data to invert the chlorophyll-a concentration in Pingzhai Reservoir. They selected the measured chlorophyll-a concentration data and quasi-synchronous Sentinel-2 data from 17 to 18 November 2017 in Pingzhai Reservoir, established a BP neural network model by selecting the optimal band combination, inverted the chlorophyll-a in Pingzhai Reservoir, and analyzed its spatial distribution characteristics. The results showed that the Sentinel-2 red-edge band was more sensitive to chlorophyll-a than the visible light band, indicating great potential for chlorophyll-a concentration inversion. Feng Tianshi et al. [10] inverted the chlorophyll-a concentration in Lake Chao based on Zhuhai-1 hyperspectral data. They extracted the remote sensing reflectance curves at the measured points from the images, screened bands with significant spectral characteristics, and used the OIF index to measure the ability of different band combinations to obtain water component information, thereby constructing a band combination with a high correlation with the measured chlorophyll-a concentration. Xu Yi et al. [11] compared four machine learning models based on measured chlorophyll-a concentration data in Lake Taihu and synchronized HJ-1B satellite CCD multispectral images. By comprehensively considering the validation accuracy, stability, and robustness of the four models, they found that the DL model had great application potential for inverting the chlorophyll-a concentration in Taihu Lake, providing a reference for studying lake water color parameters.

In terms of chlorophyll-a prediction models, they can be divided into classical models and machine learning models. Among the studies on classical models, Francisca Barraza-Moraga et al. [12] used multiple linear regression to compare the efficiency and performance of L1C and L2A products. An algorithm combining spectral bands showed a good correlation with measured Chl-a, with an R^2^ generally greater than 0.87, and RMSE and MAE errors of less than 6 µg/L. The spatial distribution of Chl-a concentration at the study site was obtained based on the proposed equation. Jiru Wang et al. [13] used the GOCI spectral remote sensing reflectance (Rrs(λ)) product to construct a quantitative model for the spatiotemporal distribution of Chl-a in the Bohai Sea and Yellow Sea areas. Three empirical ocean color algorithms and four machine learning methods were used to establish the Chl-a inversion model. The results showed that the retrieval accuracy of the machine learning methods was much higher than that of the empirical algorithms. Moses et al. used measured data from 2008 to establish a two-band model and validated the model using measured data from 2009, confirming that MERIS imagery could be used to estimate Chl-a concentrations in Case II waters in real time [14]. Nazeer et al. [15] used the third and first bands of Landsat TM and ETM+ data to invert the water quality parameter concentrations in the waters around Hong Kong, and they analyzed the dynamic changes in chlorophyll-a concentrations from 2000 to 2012. Gitelson et al. used MERIS imagery as a data source to establish a fixed three-band model, and the results showed that the constructed three-band model was suitable for different water bodies [16]. Meenu et al. used Landsat-8 imagery to establish an NIR-red algorithm to invert the chlorophyll concentration in the Dengsu River Basin [17]. Zhang Yuanzhi et al. compared semi-empirical and empirical models established based on measured hyperspectral and SAR data, and the results showed that the use of SAR data improved chlorophyll estimation [18]. Zihong Qin et al. [19] used 20 years of MODIS data to create a new empirical model to study the spatiotemporal patterns and trends of chlorophyll-a concentration in eutrophic Taihu Lake water bodies. The validation results showed that the developed model had considerable performance in estimating Chl-a, with a mean absolute percentage error (MAPE) of 12.95 µg/L and a root mean square error (RMSE) of 29.98%.

In the research on inversion models using machine learning, Sriniketan Sridhar et al. [20] acquired elevation data and chlorophyll concentration data from the Orinoco River and developed a deep-learning neural network architecture for predicting chlorophyll concentration. Xinhao Zhang et al. [21] proposed a method to construct a generic convolutional neural network (CNN) model, training and evaluating six models with different auxiliary input schemes and architectures. The results showed that the model performed better than traditional interpolation methods for unfamiliar regions, especially those outside the data coverage areas. Haibin Ye et al. [22] utilized the spectral information of MODIS/Aqua visible bands’ remote sensing reflectance and proposed a two-stage convolutional neural network (CNN) named Cchla-Net for determining chlorophyll-a concentration. Khalid A. Ali et al. [23] investigated the utility of machine learning techniques based on partial least squares (PLSs) and artificial neural networks (ANNs) in estimating low chlorophyll-a (Chl-a) concentrations in the western basin of Lake Erie (WBLE). The results showed that the PLS-ANN method could accurately estimate and monitor low concentrations of Chl-a in optically complex waters. Jianghua Ren et al. [24] proposed an improved support vector regression algorithm (DE-GWO-SVR) by introducing a hybrid differential evolution–grey wolf optimizer (DE-GWO) algorithm into the parameter selection process of the support vector regression model. This algorithm was used for the remote sensing inversion of chlorophyll-a concentration and suspended sediment concentration in the Tangdaowan Sea area. The experimental results confirmed that the DE-GWO-SVR algorithm is an effective method for the remote sensing inversion of chlorophyll-a and suspended sediment concentration in water bodies, providing a reference for the remote sensing inversion of chlorophyll-a and suspended sediment concentration in China’s offshore waters and the subsequent scientific management of water bodies. Wei-Dong Zhu et al. [25] combined hyperspectral remote sensing data and utilized a BP neural network model to invert chlorophyll-a concentration. The chlorophyll-a concentration in the study area in July and November 2020 was predicted, and the results showed that the root mean square error and average relative error between the predicted and measured values of the chlorophyll-a concentration were 2.12 µg/L and 9.66%, respectively. Qingdian Meng et al. [26] took the northern coast of the Yellow Sea as an example to investigate the performance of the enhanced spatial and temporal adaptive reflectance fusion model (ESTARFM) in fusing GOCI and Landsat Chl-a data. The results showed that both fusion models could integrate the advantages of multiple data sources to obtain Chl-a images with high spatial and temporal resolutions, thereby better realizing the monitoring of nearshore Chl-a changes.

Regarding the research on the relationship between chlorophyll-a concentration and water salinity, Hakanson Lars studied the relationship between salinity and chlorophyll-a concentration in a sea area and constructed an empirical model between salinity and chlorophyll-a concentration [27]. Hu Yi analyzed the vertical distribution characteristics and relationships among temperature, salinity, and chlorophyll-a fluorescence in the waters around the Taiwan Bank in the southern Taiwan Strait during the period of July–August 2004 based on observed data. The results showed that changes in temperature and salinity environmental factors have an important impact on the vertical distribution of chlorophyll-a fluorescence [28].

The objective of this study is to achieve the accurate monitoring of chlorophyll-a concentration in hypersaline water environments, specifically in Daihai Lake, using remote sensing technology. By comprehensively considering various environmental parameters, particularly salinity, the accuracy of the chlorophyll-a concentration prediction model is significantly improved. Salinity is one of the crucial factors affecting aquatic ecosystems, and it has a significant correlation with the content of chlorophyll-a in water bodies. Therefore, this study aims to develop and analyze various machine learning algorithms in order to identify the most suitable model for predicting chlorophyll-a concentration. By comparing the model’s accuracy before and after incorporating the salinity factor, we aim to achieve precise predictions of chlorophyll-a concentration. Through this research, we hope to provide strong scientific support for water quality monitoring and management in hypersaline water environments.

## 2. Study Area and Data

### 2.1. Study Area

Lake Daihai is located in Liangcheng County, Wulanchabu City, Inner Mongolia Autonomous Region. It is located in the southern part of the eastern section of the Yinshan Mountains and to the north of the ancient Great Wall of Yanbei. It is a typical inland closed saline lake. As shown in Figure 1, the Lake Daihai area is located between 40°29′~40°37′ north latitude and 112°33′~112°47′ east longitude. Lake Daihai was first produced by the orogenic movement in the Tertiary period. It is a common lake constructed by inland saltwater bodies. Daihai has a long oval shape and its volume is 989 million m^3^. It is the third-largest lake in Inner Mongolia [29]. Due to the withdrawal of surrounding farmland and the development of power plants, the pollution of the Dai seawater environment is serious; large-scale blooms occur every year, and the eutrophication of water bodies is high. The surface of Lake Daihai is sharply shrinking, the degree of salinization is intensifying, the water quality continues to deteriorate, and the ecological degradation is serious [30]. The lake, which is an inland, closed, brackish lake, has been closed for a long time [31]. The salinity of Lake Daihai increases year by year, and its salinity content of 15,404 mg/L makes it a high-salinity water body, resulting in a large number of fish deaths. Inner Mongolia fully promotes the comprehensive management of the “One Lake and Two Seas”, and it regards the protection and management of the ecological environment of the “One Lake and Two Seas” as an important political task [32]. The Inner Mongolia Autonomous Region has included Lake Daihai in the “One Lake and Two Seas” ecological management and has implemented the “internal governance and external introduction” project to fully control and protect the ecological environment of Daihai Lake. The “Yellow River reclamation” project began in early 2022.

### 2.2. Data

#### 2.2.1. Remote Sensing Data Preprocessing

In the field of optical remote sensing, an imaging spectrometer simultaneously images ground objects in the visible light to the thermal infrared bands, producing hyperspectral image data and recording the spatial information, radiation information, and spectral curve of ground objects [33]. In order to achieve real-time and accurate monitoring of inland lake water bodies, it is necessary to use hyperspectral remote sensing image data within a short revisit period and a detailed description of surface features. A hyperspectral satellite developed by Obit Inc. was successfully launched in 2018. To date, many water quality parameters of lake water environments have been retrieved [34]. Remote sensing monitoring and evaluation of water quality indicators have been realized. Zhuhai Obit Aerospace Technology Co., Ltd. (Zhuhai, China) successfully launched four Zhuhai No.1 hyperspectral satellites on 26 April 2018, and another four on 19 September 2019 [35]. The spatial resolution of the hyperspectral data of Zhuhai-1 is 10 m, the average spectral resolution is 2.5 nm, there are 32 bands, and the spectral range is 400–1000 nm. The main parameter settings of the Zhuhai-1 hyperspectral satellite are shown in Table 1.

The remote sensing data selected in this study were obtained from an image taken by the Zhuhai No.1 hyperspectral satellite when passing through Daihai on 6 September 2023. The bands used in the chlorophyll-a concentration inversion method were obtained from the Zhuhai No.1 data. The band setting of the Zhuhai No.1 hyperspectral image used in this study is shown in Table 2. It provides reliable image data for the quantitative inversion of chlorophyll-a in water bodies at the spatial scale. Because the remote sensing image is affected by the atmospheric radiation transmission process between the water body and the sensor, the image needs to be preprocessed accordingly.

Remote sensing data preprocessing:Image reading

In this study, ENVI5.3 software was used to read the hyperspectral image of Zhuhai No.1. After reading the Euclidean hyperspectral image, field data such as the center wavelength, the half-height width, and the calibration coefficient were automatically integrated and the RPC information was identified. The image source file contains information on 32 bands. When reading the image for the first time, the * _ B15 _ * meta.xml file needs to be opened to automatically combine the 32 tiff file bands.

2.Radiative calibration

Radiometric calibration is a necessary processing step before atmospheric correction. The gray value of the pixel record in the image is converted into the absolute radiance value (emissivity) via radiometric calibration processing. In this study, the Radiometric Calibration tool in ENVI5.3 software was used for processing. The radiation calibration formula used in the orbit hyperspectral image is
(1)$$Le=gain\times \frac{DN}{TDIStage}+offset$$

In the formula, Le is the radiation brightness value, gain is the gain coefficient, DN is the pixel gray value of the original image record, TDIStage is the integral series, and offset is the absolute radiometric calibration offset coefficient.

3.Atmospheric correction

The spectral response of the sensor to the water body is low and susceptible to the atmosphere. The optical radiation signals received by the remote sensing platform include many radiation signals related and unrelated to water quality parameters. In order to obtain the true reflectivity of ground objects, it is necessary to remove the radiation signal reflected by non-lake water bodies from the total radiation signal. Among them, only the radiation brightness from the water carries the water body information, accounting for only a small part of the image information, and the rest is interference information. In order to improve the inversion accuracy of water quality parameters as much as possible, this study compared a variety of atmospheric correction algorithms, and, finally, the fastest atmospheric correction algorithm was used to perform atmospheric correction on the Zhuhai-1 hyperspectral image. The radiation calibration result of the image was used as the input image of the atmospheric correction, and the image type was set as Hyperion to select the image output path. After waiting for the software to process for a period of time, the image data processed by the atmospheric correction were obtained.

A vegetation pixel of the hyperspectral image was selected; the spectral curves extracted from the pixel before and after atmospheric correction are shown in Figure 2. It can be seen that the spectral curve before atmospheric correction has a high reflectivity in the visible light band, and there are reflection peaks at 480 nm and 550 nm. The reflectivity near 520 nm is low due to the absorption of chlorophyll, and there is an obvious reflection valley at 760 nm. After 800 nm, the reflectivity curve shows a downward trend. After atmospheric correction, the spectral curve of the vegetation pixels is within 700 nm due to the absorption of chlorophyll, carotenoids, and the atmosphere, resulting in low reflectance. After 700 nm, the reflectance spectral curve rises sharply. Due to the high reflection of infrared light by vegetation, the spectral curve has an upward trend. Through a comparative analysis of the vegetation pixels before and after atmospheric correction processing, it is concluded that the spectral curve after atmospheric correction has the actual measured vegetation spectral characteristics, which confirms the effectiveness of atmospheric correction processing in this study.

4.Orthocorrection

Orthorectification is when the RPC information of an image file is used to correct the spatial and geometric distortions generated during the imaging process, and the correct geographical location is obtained through orthorectification so that the image matches the actual ground objects. In this study, the digital elevation model (DEM) was used as auxiliary data to orthorectify the hyperspectral image of Zhuhai No.1. The orthographic correction file (_rpc.txt) is provided in the data file of the Zhuhai No.1 hyperspectral image, and the process tool RPC Orthorectification Workflow is used for the processing. In orthorectification, the output pixel size is set to 10 m, and the resampling method uses three convolutions to automatically determine the offset of ground objects. Through the processing of orthorectification, the ground objects in the image correspond to the actual geographical location, which increases the accuracy of the inversion results.

#### 2.2.2. Field Data

When a water bloom breaks out, the chlorophyll concentration in most areas of the water body generally exceeds 100 mg·m^−3^. At this time, the concentration of chlorophyll-a is used as an early warning of the water bloom. On the eve or in the early stage of a water bloom, monitoring the concentration of chlorophyll-a in the water body can give the biomass and water quality of the algae in the water body, which is of great significance for the early warning and targeted prevention of water blooms. After mastering the transit date and revisit period of the satellite, the field sampling date was determined to be the morning of 6 September 2023. The weather was sunny, the amount of cloud was low, and the influence of aerosol on water reflection was small. Zhuhai-1 satellite data were obtained in the same time period. In the fieldwork experiment area, 30 sampling points were evenly set up using the GPS satellite locator in the determined area (the position is shown in Figure 3). Using a collector, lake water samples were collected at each sampling point. A pre-prepared magnesium carbonate solution was added, and the samples were stored in a dark place before being sent to the laboratory for the spectrophotometric measurement of chlorophyll-a concentration. Meanwhile, the salinity data for the corresponding sampling points were tested using the conductivity method. Finally, 30 valid samples of chlorophyll-a concentration and 30 valid samples of salinity were obtained.

## 3. Methods

### 3.1. Overview of Methods

Combined with the remote sensing data and the measured data from the sampling points, the research steps for this study are shown in Figure 4.

This article first clarifies the experimental research area and describes the preprocessing of the simultaneously collected measurement data for the sampling points and hyperspectral imagery data from the Zhuhai-1 satellite. The Zhuhai-1 hyperspectral imagery underwent preprocessing procedures such as radiometric calibration and atmospheric correction to obtain reflectance remote sensing imagery. Based on the coordinate information for the sampling points, band reflectance values corresponding to the sampling points in the imagery were extracted.

Initially, without considering the impact of salinity, correlation analysis and principal component analysis (PCA) dimensionality reduction were performed on the preprocessed Zhuhai-1 satellite data to screen out key characteristic bands. The sampling point data were divided into a training set and a test set, and a machine learning model was employed for training. After comparing the precision performance of different models on the same data, we ultimately selected the random forest model with the highest accuracy for training. Subsequently, the salinity factor was incorporated into the random forest model, and a PCA was re-performed to obtain new characteristic bands, further evaluating the inversion accuracy of the model. Finally, cross-validation was performed on the established model, and a comparison of the model’s precision before and after the addition of the salinity factor was made. Through this step, an in-depth analysis of the influence of salinity on the inversion of chlorophyll-a concentration was conducted, thus providing a more precise scientific basis for water quality monitoring and management in high-salinity water environments.

### 3.2. Principal Component Analysis

#### 3.2.1. Principle

Principal component analysis attempts to recombine many original indicators with a certain correlation (such as p indicators) into a new set of unrelated comprehensive indicators, thereby replacing the original indicators. Its geometric meaning is to re-represent the original space with a new coordinate system. The usual treatment is to make a linear combination of the original p indicators as a new comprehensive indicator [36]. The principal component analysis (PCA) is a multivariate data analysis method that aims to reduce the dimension of data while retaining as much information as possible. It is widely used in data preprocessing, feature selection, data visualization, and other fields.

#### 3.2.2. Mathematical Model

(1)Calculate correlation coefficient matrix p

(2)$$R=\left[\begin{array}{c} R_{11}\;R_{12}\ldots R_{1p}\\ R_{21}\;R_{22}\ldots R_{2p}\\ \vdots \;\;\;\;\;\;\;\vdots \;\;\;\;\;\vdots \;\;\;\;\vdots \\ R_{p1}\;R_{p2}\ldots R_{PP} \end{array}\right]$$
where $R_{ij}$ (i, j = 1, 2, …, p) is the correlation coefficient between the original variables $x_{i}$ and $x_{j}$.

(2)Calculate eigenvalues and eigenvectors

First, solve the characteristic equation $\left|\lambda I-R\right|$ = 0, typically employing the Jacobi method, to obtain the eigenvalues $\lambda_{i}\;(i$ = 1, 2 …, p), and arrange them in descending order $\lambda_{1}\geq \lambda_{2}\geq \cdots \geq \lambda_{p}\geq 0$. Next, determine the corresponding eigenvectors $e_{i}\;(i$ = 1, 2 …, p) associated with the eigenvalues $\lambda_{i}$, ensuring that $e_{i}=1$.

(3)Calculate the contribution rate of principal components and the cumulative contribution rate

The contribution rate of the principal component $F_{i}$ is
(3)$$\frac{\lambda_{i}}{\sum_{k=1}^{p}\lambda_{k}}\left(i=1,2\ldots p\right)$$

The cumulative contribution rate is
(4)$$\frac{\sum_{k=1}^{i}\lambda_{k}}{\sum_{k=1}^{p}\lambda_{k}}\left(i=1,2\ldots p\right)$$

Generally, the first, second, …, $m\left(m\leq p\right)$ principal components corresponding to the eigenvalues $\lambda_{1},\cdots \lambda_{2},\cdots ,\lambda_{m}$ with a cumulative contribution rate of 85–95% are taken.

In this study, a principal component analysis is carried out on the 32 bands of the Zhuhai-1 hyperspectral image of the Daihai area after preprocessing. The band reflectance value obtained by preprocessing the hyperspectral images represents the absorption and scattering of optical signals by water bodies at the sampling points. Therefore, it is necessary to screen out the band set that contains the original band information, as this can further simplify the chlorophyll-a concentration inversion analysis process.

### 3.3. Cross-Validation

Cross-validation is a model evaluation technique commonly used in machine learning research. It accurately estimates model performance by dividing datasets, training, and testing multiple times. K-fold cross-validation is a typical practice. Its core idea is to divide the dataset into *k* independent subsets (fold), with each subset used as the test set in turn, and the remaining *k* − 1 subsets used as the training set; the process is repeated *k* times. Finally, the performance indices of *k* experiments are averaged to obtain a more robust evaluation of the model performance. The process of K-fold cross-validation is as follows:(1)The dataset is divided into *k* subsets: $D=\left\{D_{1},D_{2},\ldots ,D_{k}\right\}$.(2)For each i, where i=1, 2,…,k:Let $D_{test}=D_{i}$ be the test set.Let $D_{train}=D-D_{i}$ be the training set.Train models on $D_{train}$.Evaluate model performance on $D_{test}$.(3)Calculate the average value of the performance index:

(5)$$Performance=\frac{1}{k}\sum_{i=1}^{k}Performance_{i}$$
where $Performance_{i}$ is the performance index of the *i*-th experiment.

### 3.4. Correlation Analysis

The methods for measuring seawater salinity mainly include chemical methods, optical methods, density methods, conductivity methods, microwave remote sensing methods, and measurement methods based on fiber optic technology. Of these, the conductivity method is a relatively fast, accurate, and suitable method for laboratory measurements. Its basic principle involves measuring the conductivity of seawater to calculate its salinity [37]. Thirty collected chlorophyll-a values were analyzed in relation to salinity values. As shown in Figure 5, there is a positive correlation between salinity and chlorophyll-a concentration in Lake Daihai, where the chlorophyll-a concentration increases with the rise in salinity. This shows that it is necessary to consider the salinity factor when inverting the chlorophyll-a concentration of Lake Daihai, a high-salinity water body.

### 3.5. Machine Learning Model

#### 3.5.1. Random Forest Model

Random forest performs well in dealing with high-dimensional data, and it also has good robustness for missing and unbalanced data [38]. As shown in Figure 6, Due to the independence of each tree, the training process can be easily parallelized to improve efficiency. Random forests usually show a strong generalization ability for new data, and they can maintain stable performance, even in the face of noisy data or complex problems [39]. Due to their excellent performance on various types of datasets, random forests are still widely welcomed and applied in the field of machine learning.

#### 3.5.2. Ensemble Learning—Model Stacking

Ensemble learning is a machine learning approach that combines multiple models to improve prediction accuracy and stability. The fundamental idea behind this method is to integrate multiple weak learners (i.e., models that do not perform exceptionally well individually) to form a more powerful and reliable model. Model stacking is a type of ensemble model that utilizes the output of multiple component models as input to train a new machine learning model [40]. As shown in Figure 7, the entire training dataset is first divided into multiple subsets through resampling methods, and these newly generated training subsets are then used to independently train a series of classification models, referred to as Tier 1. The outputs of these Tier 1 classifiers are then combined and used as input to train the Tier 2 meta-classifier. In the process of training Tier 1 classifiers, besides resampling methods, cross-validation strategies are often employed. Specifically, the training set is first evenly divided into N parts, and then each individual learner in Tier 1 is trained on the first N − 1 parts and tested on the Nth part. This approach ensures that each learner is evaluated on unseen data, thus improving the model’s generalization ability.

When applying ensemble learning to invert chlorophyll-a concentration, integrating different models can reduce the risk of overfitting because different models tend to perform better on different subsets of data. This diversity contributes to improving the overall prediction accuracy. Ensemble learning enhances the generalization ability to new data by combining multiple models, which is particularly crucial when predicting chlorophyll-a concentrations that vary significantly in the environment. Overall, ensemble learning provides an accurate and stable approach for inverting chlorophyll-a concentration, and it is especially suitable for handling complex data issues in environmental monitoring and ecological research. Model stacking, a technique within ensemble learning, involves combining multiple different base learners (base models) to improve prediction accuracy. The key feature of the stacking method is that it uses a new model to integrate the predictions of the individual base learners. This approach ensures that the meta-learner does not simply replicate the behavior of the base learners but instead truly learns how to optimally combine their predictions.

#### 3.5.3. Ridge Regression Model

Ridge regression is an enhanced linear regression technique that addresses the common issue of multicollinearity in ordinary linear regression. Its core feature is the introduction of an L2 regularization term, which is a penalty on the sum of squared model coefficients. By controlling the magnitude of model parameters, it effectively mitigates the instability caused by highly correlated predictor variables. The primary functions of this method include reducing model complexity and improving robustness to input data, especially in cases where high collinearity exists in the dataset. Collinearity can lead to unstable estimates of model parameters, and ridge regression finds a balance between fitting the model and preventing overfitting through the regularization term.

The benefits of ridge regression are demonstrated not only in its effective handling of collinearity but also in its resilience to noise. By adjusting the regularization parameter, researchers can flexibly control the complexity of the model to optimize its performance. In practical operation, methods such as cross-validation are often used to select the optimal regularization parameter to ensure the model’s generalization ability across different datasets.

Overall, ridge regression exhibits significant advantages in handling practical linear regression problems, providing researchers with a reliable tool, especially when faced with complex data structures and high-dimensional data. The advantages of ridge regression models in inverting water chlorophyll concentration mainly lie in their ability to effectively handle multicollinearity, prevent overfitting, and adapt to incomplete feature sets in water remote sensing data. They also possess strong interpretability and flexibility, making them widely adopted in practical applications.

## 4. Results and Discussion

### 4.1. Spectral Characteristics Analysis

The correlation analysis between the spectral curve extracted from the hyperspectral image corresponding to the sampling point and the measured chlorophyll-a concentration is shown in Figure 8. In the map, it can be observed that the remote sensing reflectance curve of Lake Daihai shows the characteristics of a typical second-class water body, that is, a saline water body. The B0–B5 band is 490 nm–550 nm; this range covers part of the blue-to-green region. Chlorophyll-a has an absorption peak in the blue region, so lower reflectivity is seen at around 490 nm. In the green band, chlorophyll absorption decreases, so the reflectivity increases. The B5–B10 band is 560 nm–640 nm; this region spans the green-to-red region. The high reflectance in the green region gradually decreases as chlorophyll begins to show stronger absorption near 640 nm. The B10–B15 band is 665 nm–709 nm; this interval contains the red-light region in which chlorophyll-a has a significant absorption peak at about 670 nm. In this band, the reflectance decreases significantly, and this absorption peak can be used to estimate the concentration of chlorophyll-a. The B15–B20 band is 730 nm–780 nm; this band is located in the red-edge region, that is, from the red-light region with more chlorophyll absorption to the near-infrared region with less absorption. The position and slope of the red edge are usually related to the chlorophyll content and can be used to monitor the health status of vegetation. The B20–B25 band is 806 nm–865 nm. In this near-infrared region, the reflectance of plants and water is usually higher. Although the absorption of chlorophyll in this area is low, the pattern of reflectance can help distinguish vegetation types and densities. The B25–B30 band is 880 nm–940 nm; this range continues to expand the near-infrared region, which is often used to estimate vegetation biomass.

The spectral curve shows the changes in the observation data of the Zhuhai-1 satellite under different bands. The changes in specific bands can reveal the concentration of chlorophyll-a and the possible health status of water and vegetation. In practical applications, this information is of great value for environmental monitoring, agricultural production, and water quality analysis. In water remote sensing, these bands can help estimate the content of organic matter and suspended matter in water, and they indirectly affect the assessment of chlorophyll-a concentration.

### 4.2. Characteristic Band Analysis

Through an in-depth analysis of the principal component analysis method, the characteristic bands required by the hindsight model are successfully obtained (the detailed data are shown in Table 3). Among the many principal components, the first eight principal components are carefully selected. The information carried by these principal components is sufficient to cover up to 98.3% of the variance information in the original dataset, which fully demonstrates their high efficiency and accuracy in data expression.

Further, through a detailed comparison, the band information used by the model before and after the addition of the salinity factor is observed. In this process, the six bands of B4, B13, B21, B24, B27, and B29 are effectively utilized by the model in both cases. This finding not only proves that these bands have a very high contribution to the inversion of chlorophyll-a concentration but also reveals the strong correlation between them and chlorophyll-a concentration. It provides key information for the inversion of chlorophyll-a concentration, and its importance in data analysis cannot be ignored.

The correlation coefficient between the principal component band and the measured chlorophyll-a concentration is shown in Figure 9. In this figure, it can be intuitively observed that the bands with large correlation coefficients with chlorophyll-a concentration are mainly B21 and B27. Specifically, the correlation coefficient of the B27 band is as high as 0.2874, showing a close correlation with chlorophyll-a concentration. At the same time, the correlation between the B21 band and the measured value is also relatively high, but it shows a negative correlation, with a correlation coefficient of −0.2231. This means that the numerical change in the B21 band is opposite to the change direction of chlorophyll-a concentration. Although its correlation is slightly lower than that of the B27 band, it is still an important factor that cannot be ignored. This finding has important guiding significance for understanding the variation in chlorophyll-a concentration and optimizing the relevant inversion model.

### 4.3. No Salinity Data

First, the 30 sampling points were divided into a training set and a testing set, with 80% of the data used for model training and the remaining 20% used for validating the model’s performance. After conducting band selection and analysis of the Zhuhai-1 remote sensing imagery combined with the spectral curve characteristics of the sampling points, machine learning models based on hyperspectral remote sensing were constructed. These models included a random forest model, the model stacking approach from ensemble learning, and a ridge regression model.

Figure 10 shows the final prediction results of these three machine learning models for the target variable. In this initial stage, without considering the influence of salinity factors, it was found through comparison that the different models exhibited varying prediction accuracies after training on the same dataset.

Specifically, Figure 10a represents the model stacking approach in ensemble learning, with a root mean square error (RMSE) of 0.109 and a coefficient of determination (R^2^) of 0.52. Figure 10b depicts the ridge regression model, with an RMSE of 0.084 and an R^2^ of 0.61. Figure 10c shows the performance of the random forest model in inverting chlorophyll-a concentration, achieving an RMSE of only 0.056 and an R^2^ of 0.64.

By comparing the predictive performance of these three models, it is evident that the random forest model significantly outperforms the other two models in terms of prediction accuracy, both in terms of the RMSE and R^2^. Therefore, the random forest model is adopted to further train and analyze the data from the Zhuhai-1 satellite in order to obtain more accurate and reliable prediction results.

After an in-depth analysis of the data from the Zhuhai-1 satellite combined with the inversion results of the random forest model, as shown in Figure 11, it is found that, in the specific area studied, the concentration of chlorophyll-a in the northern part of Lake Daihai is higher, indicating a high accumulation of algae biomass in the water body, which is related to an increase in nutrient levels in the water body. In contrast, the concentration of chlorophyll-a in the northeastern and southern lake areas is relatively low. It is worth noting that there is a large gap between the retrieved chlorophyll-a concentration value and the actual value, thus further in-depth study and optimization of the model is required to improve the accuracy of the inversion.

### 4.4. Addition of Salinity Factor

In order to further improve the prediction accuracy of the model, the previously constructed random forest model was further optimized; in particular, salinity was included as a crucial variable in the model. After this careful adjustment, a new prediction result was obtained; the detailed data are presented in Table 4. By carefully examining the data in the table, it was intuitively found that the prediction performance of the model showed a significant improvement after the introduction of the salinity factor. This improvement not only enhanced the model’s ability to fit the actual situation but also provided more solid and reliable data support for subsequent research work.

In order to further verify the effect of this improvement, a result verification diagram of each model after the addition of the salinity factor was drawn, as shown in Figure 12. It could be clearly observed that, after introducing the salinity factor, the prediction accuracy of the model significantly improved. Specifically, the root mean square error (RMSE) was 0.047 and the coefficient of determination (R^2^) was 0.92. This indicates that the prediction accuracy of the model improved after salinity data were added to it. The coefficient of determination (R^2^) continued to approach 1, which indicates that the fitting degree between the predicted value and the actual value of the model was higher and the prediction result was more reliable. At the same time, the root mean square error (RMSE) continued to decrease, which further proves the improvement of model prediction accuracy.

This optimization makes the model more accurate and reliable in predicting the concentration of chlorophyll-a in Lake Daihai and provides strong support for water quality monitoring and evaluation. By continuously introducing key variables and optimizing the model, the prediction accuracy can be continuously improved to provide more accurate data support for environmental protection and water resources management.

Figure 13 clearly reveals a key finding: after considering the key factor of salinity, the distribution pattern of chlorophyll-a concentration in the study area changes significantly. In particular, the previous significant high concentration phenomenon in the central region is alleviated, becoming a medium-to-high concentration, while the high concentration phenomenon in the northern marginal zone still exists, and the concentration in the southern and eastern regions is significantly reduced. This change underscores the important role of salinity in the distribution of chlorophyll-a. An in-depth analysis shows that the predicted results of the model incorporating salinity parameters are highly consistent with the actual observations, which verifies the significant improvement in the accuracy of the model.

In a high-salinity water environment, the effect of salinity on chlorophyll-a concentration is particularly important. Salinity is not only closely related to other environmental parameters of water but also interacts with biological activities, resulting in complex interrelationships in the distribution of chlorophyll-a concentration. By incorporating the variable of salinity, a model can capture these subtle correlations more accurately and thus improve the prediction accuracy. In addition, as an additional environmental indicator, salinity brings more explanatory information to a model, which makes it possible to better understand the mechanism behind the change in chlorophyll-a concentration, which enhances not only the predictive ability of the model but also its application value in scientific research.

The results of the model cross-validation are shown in Table 5.

A total of 30 datasets with sampling points were used to construct a random forest model and evaluate its predictive performance. In order to further explore the impact of salinity factors on the predictive ability of the model, one dataset included salinity factors, while the other did not.

For both datasets, a strict 5-fold cross-validation method was used to evaluate the stability and generalization ability of the model. This method ensures the fairness and accuracy of model evaluation, as it fully considers the diversity and complexity of different samples in the dataset. After a detailed model training and validation process, key indicators were obtained. The average R^2^ value of the dataset with the added salinity factors under the random forest model was 0.89, while the average R^2^ value of the dataset without the added salinity data was 0.61. This result significantly indicates the importance of salinity factors in improving model prediction accuracy, especially on such datasets. Therefore, in future data analysis and modeling processes, considering the influence of salinity factors will be an aspect that cannot be ignored.

### 4.5. Determination of Chlorophyll-a Concentration in Lake Daihai

By using the data from the Zhuhai-1 satellite and the random forest model optimized using the salinity factor, the chlorophyll-a concentration of the whole Lake Daihai was successfully inverted, as shown in Figure 14. The distribution of chlorophyll-a concentration in Lake Daihai waters can be clearly observed in the diagram. The concentration of chlorophyll-a in the coastal area was significantly higher than in the center of the lake, especially in the southern waters, where the concentration of chlorophyll-a was generally higher. This is mainly due to the Daihai power plant located on the south bank of the lake, about 600 m from the lake. The thermal discharge of the power plant has a profound impact on the physical and chemical properties of the Daihai water body and the species, quantity, community structure, and ecological environment of aquatic organisms. Thermal discharge leads to an increase in temperature in Daihai waters, which, in turn, reduces the concentration of dissolved oxygen. For every 6~10 °C increase in water temperature, dissolved oxygen decreases by about 0.5~2.0 mg, which adversely affects aquatic organisms. The increase in water temperature promotes an increase in nutrient content, thus accelerating the growth and reproduction of algae. Excessive algae blooms lead to an increase in the chlorophyll-a concentration in Daihai waters. The concentration of chlorophyll-a in the northeastern waters is significantly lower. This is due to the intermittent surface runoff, underground inflow runoff, and lake precipitation recharge in the northeastern region of the basin. These water flows dilute the water resources entering the lake, making the distribution of chlorophyll-a concentration more uniform and not concentrated at the lake entrance. In particular, Tiancheng River, Buliang River, Wuhao River, and other rivers are located in the northeastern part of the Daihai Basin, and they play a dilution role in water quality. In addition, the concentration of chlorophyll-a in the northwest coastal waters is also high. This is mainly due to the establishment of Daihai tourist attractions. A large amount of manmade pollution has seriously damaged the ecological environment of the region, resulting in increased eutrophication of the water body, which, in turn, increases the concentration of chlorophyll-a.

Through an experimental analysis, it is found that the distribution of chlorophyll-a concentration in Daihai waters is complex; in particular, the distribution of chlorophyll-a concentration in the coastal waters of Daihai Lake is higher because of the complex ecological changes around Daihai waters.

## 5. Conclusions

This study focused on hypersaline water bodies and delved into the inversion methods for chlorophyll-a concentration through the integration and application of remote sensing technology. During the research, a random forest inversion model based on Zhuhai-1 remote sensing data was successfully established, and the inversion accuracy was significantly improved by introducing the salinity factor. The main achievements of this study are as follows:

(1) This study utilized the Python language to develop an automatic principal component analysis method for band dimensionality reduction. Through this method, principal component bands with strong correlations between the Zhuhai-1 satellite data source and chlorophyll-a concentration were selected as characteristic bands to construct an inversion model for chlorophyll-a concentration. This provides crucial information for establishing high-precision remote sensing inversion models.

(2) The importance of the salinity factor in constructing an inversion model for chlorophyll-a concentration in hypersaline lakes such as Daihai was clarified, and the reliability of adding the salinity factor was verified through specific measured data. By comparing the model performance before and after the addition of the salinity factor to the same model, it was found that the inversion accuracy was significantly improved after introducing the salinity factor. This discovery is of great significance for the inversion of chlorophyll-a concentration in hypersaline water bodies.

(3) An optimal model for inverting the chlorophyll-a concentration in Daihai water bodies using satellite remote sensing data was established. Based on Zhuhai-1 satellite data, a random forest inversion model using machine learning algorithms was constructed. Through a comparative analysis of prediction results and errors, it was found that the random forest regression model with the salinity factor and data processing method developed in this study performed exceptionally well in inverting the chlorophyll-a concentration in Daihai water bodies. The determination coefficient (R^2^) of this model was as high as 0.92, and the root mean square error (RMSE) was only 0.047, indicating that the model can accurately invert the chlorophyll-a concentration in Daihai. Therefore, the random forest model with the salinity factor has a high application value in inverting the chlorophyll-a concentration in Daihai water bodies. This model not only provides high-precision technical support for the long-term and large-scale monitoring of water quality in Daihai but also lays a solid foundation for the application of remote sensing technology in hypersaline water environments.

In summary, this study provides effective methods and technical support for the remote sensing inversion of chlorophyll-a concentration in hypersaline water bodies, offering new ideas for research and practice in water quality monitoring and assessment.

## Figures and Tables

**Figure 1 sensors-24-04181-f001:**
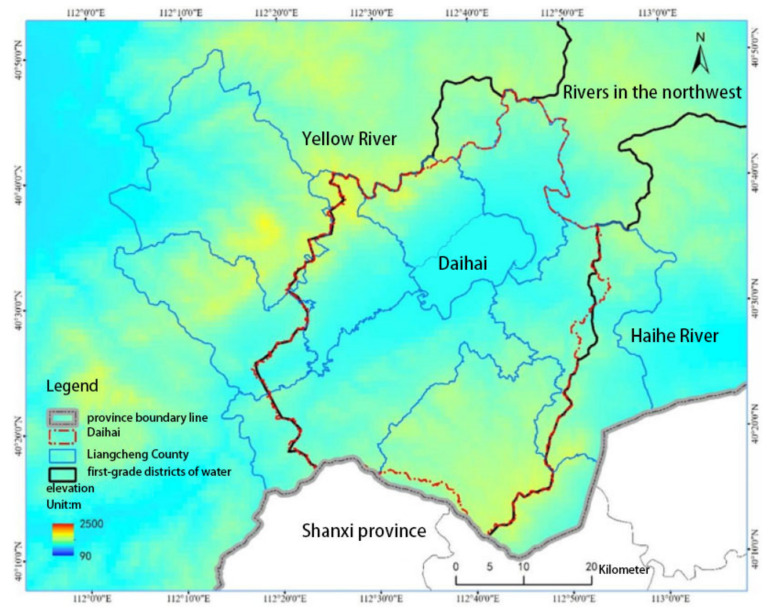
Geographical location of Daihai.

**Figure 2 sensors-24-04181-f002:**
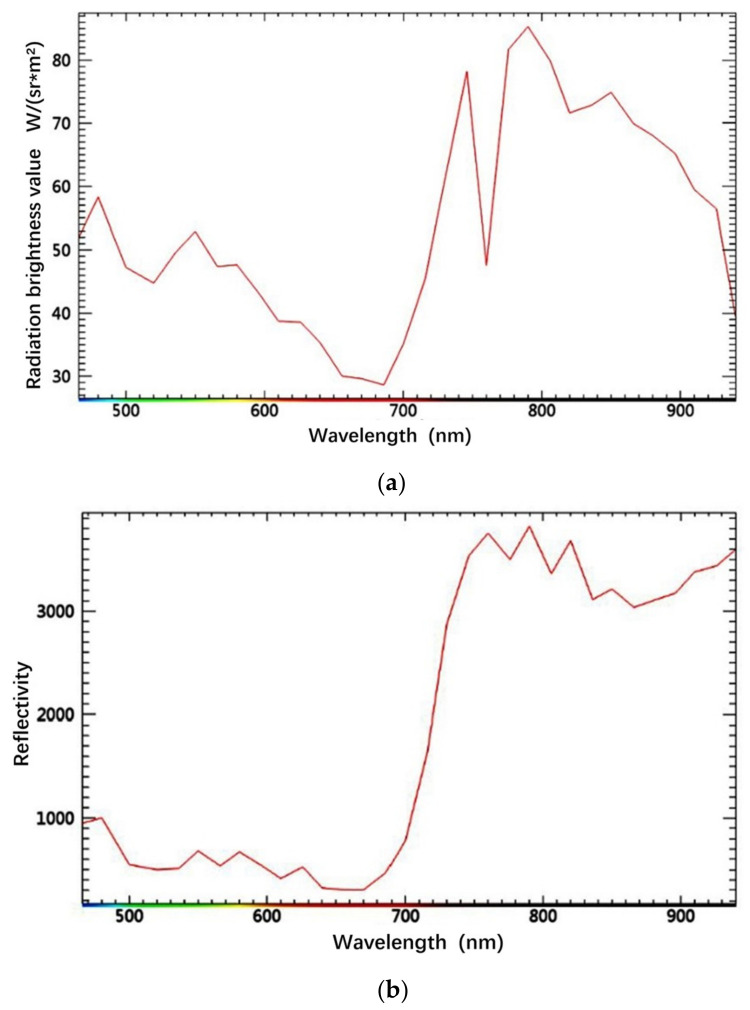
Comparison of spectral curves of vegetation pixels: (**a**) before atmospheric correction; (**b**) after atmospheric correction.

**Figure 3 sensors-24-04181-f003:**
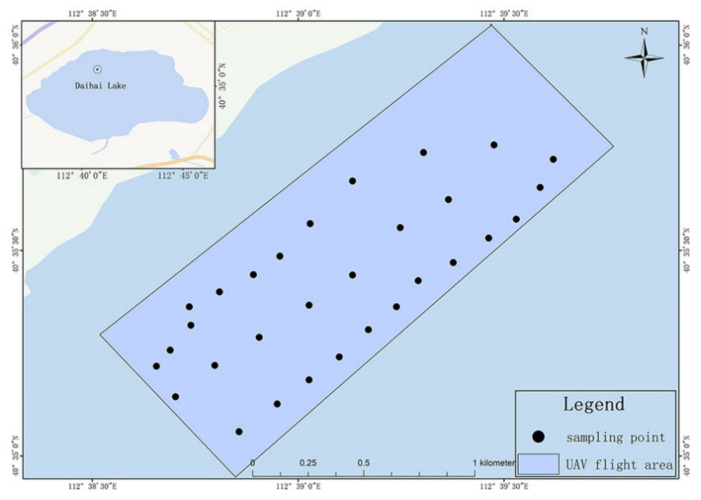
Sampling point diagram.

**Figure 4 sensors-24-04181-f004:**
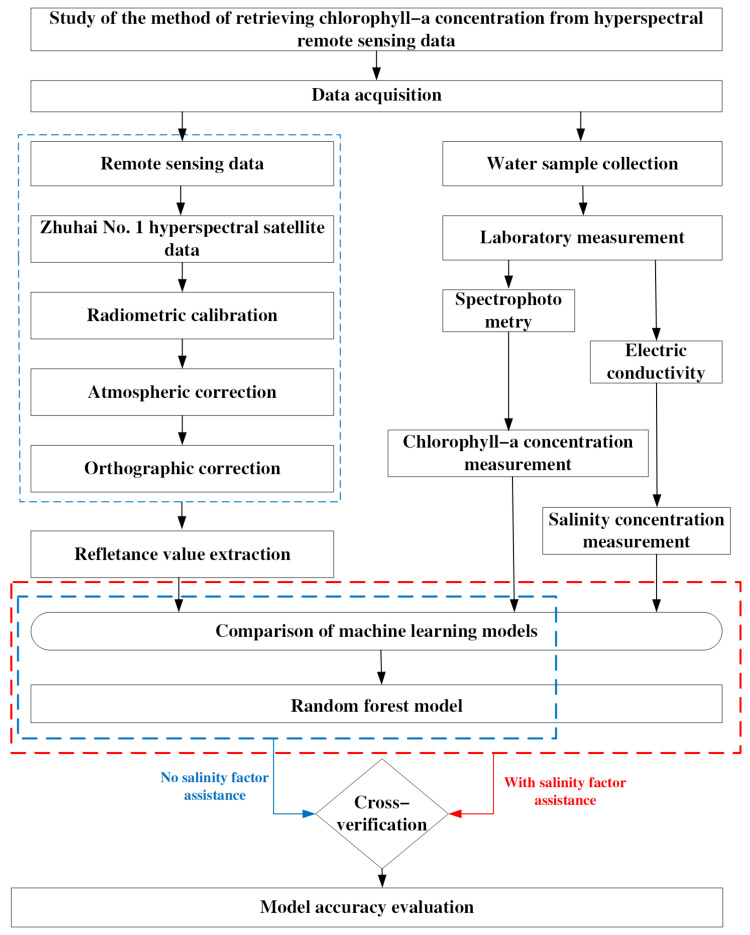
Technology flowchart.

**Figure 5 sensors-24-04181-f005:**
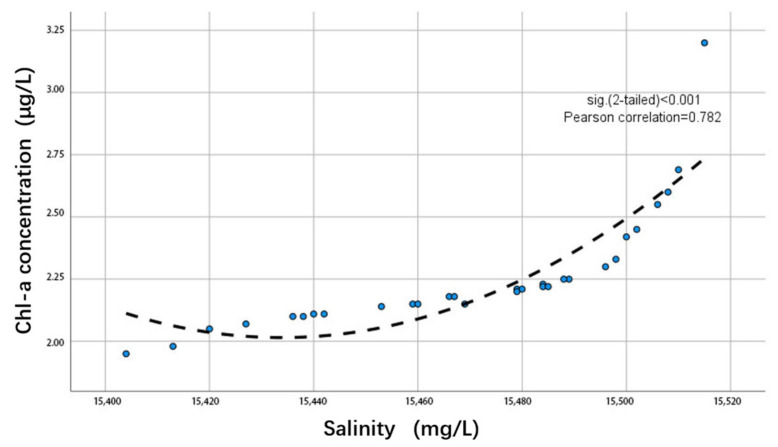
Relationship between salinity and chlorophyll-a concentration.

**Figure 6 sensors-24-04181-f006:**
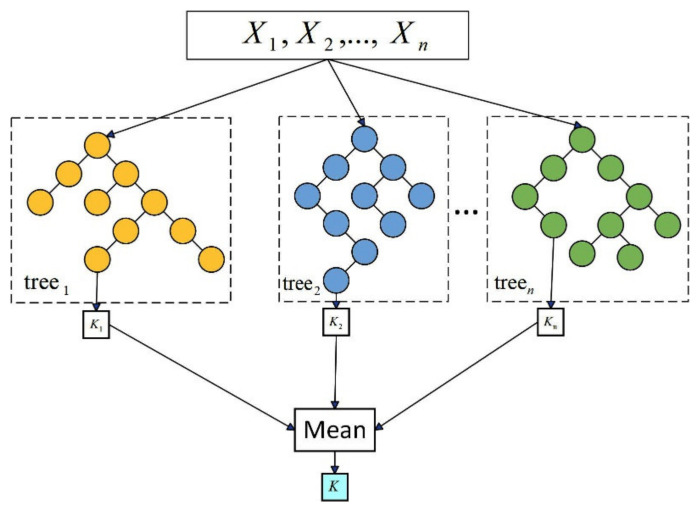
Random forest model diagram.

**Figure 7 sensors-24-04181-f007:**
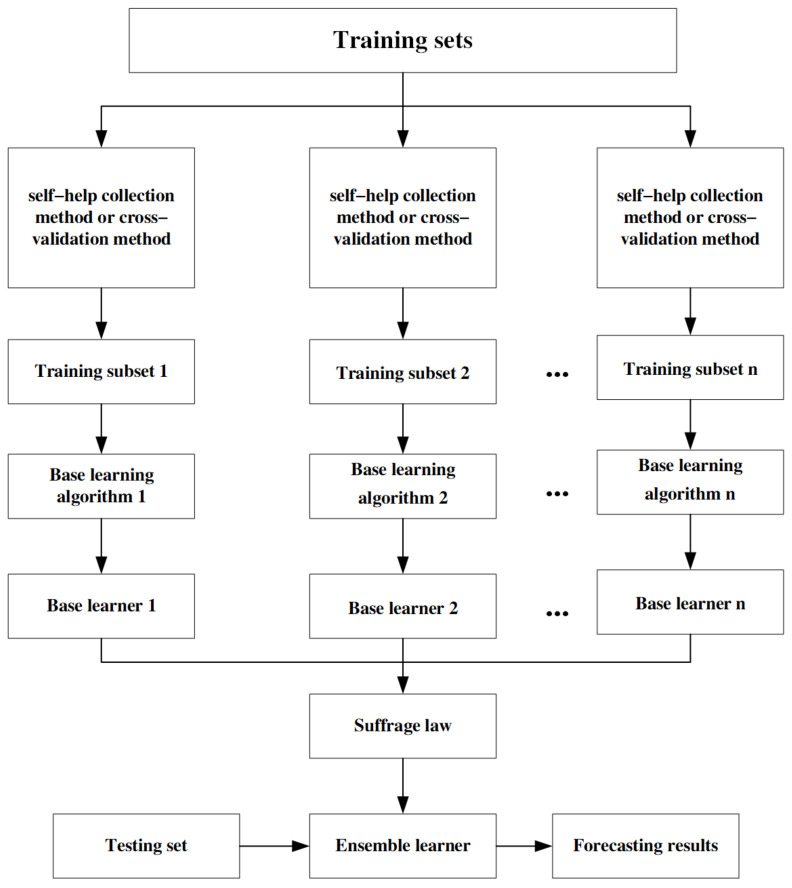
Algorithm flowchart.

**Figure 8 sensors-24-04181-f008:**
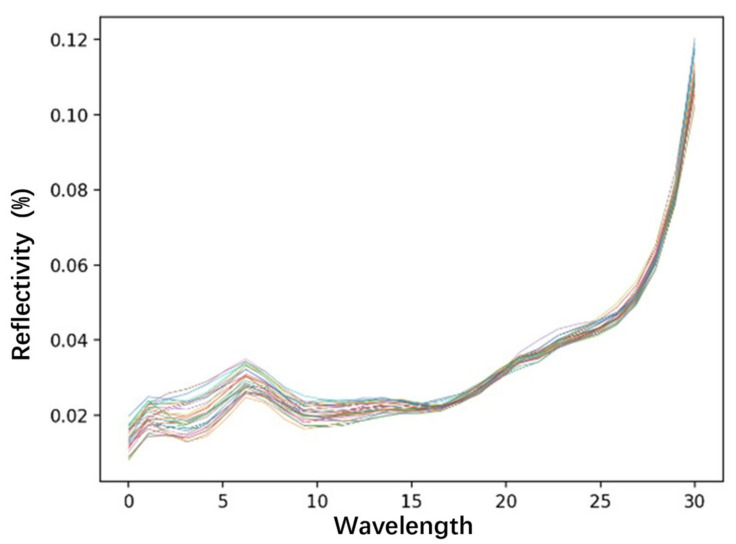
Remote sensing reflectance curve of Daihai sampling point.

**Figure 9 sensors-24-04181-f009:**
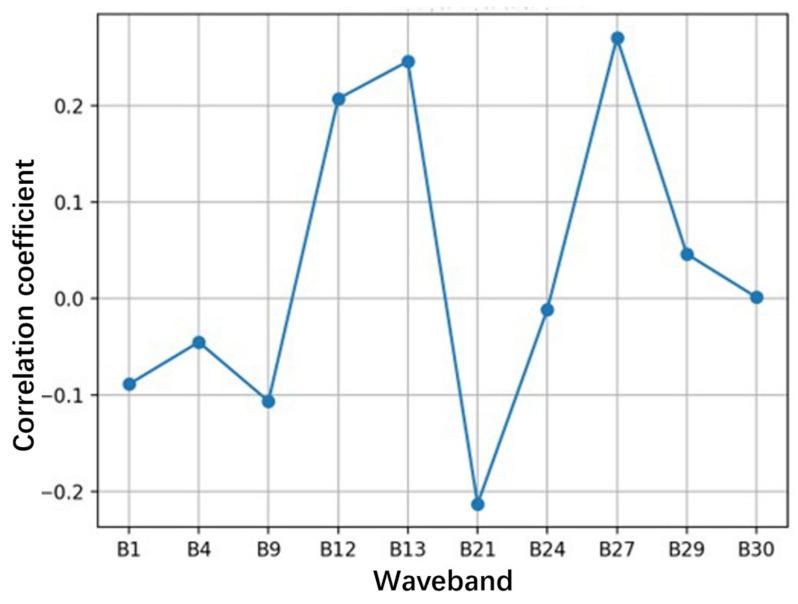
Correlation analysis of satellite characteristic band.

**Figure 10 sensors-24-04181-f010:**
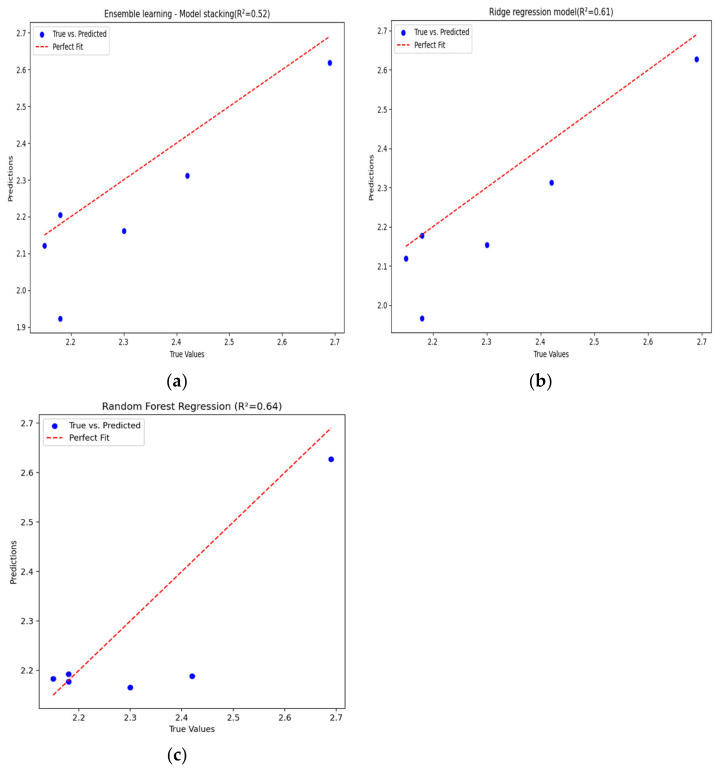
Chlorophyll-a concentration inversion model diagram (without salinity factor): (**a**) ensemble learning—model stacking; (**b**) ridge regression model; (**c**) random forest model.

**Figure 11 sensors-24-04181-f011:**
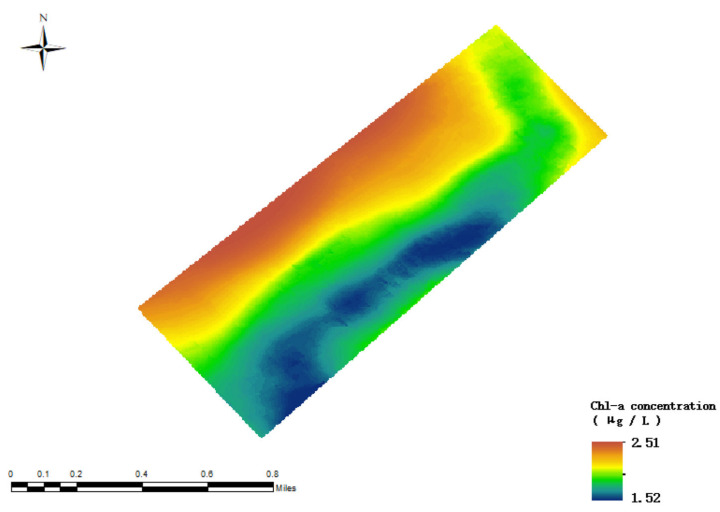
Random forest model inversion results (without addition of salinity factor).

**Figure 12 sensors-24-04181-f012:**
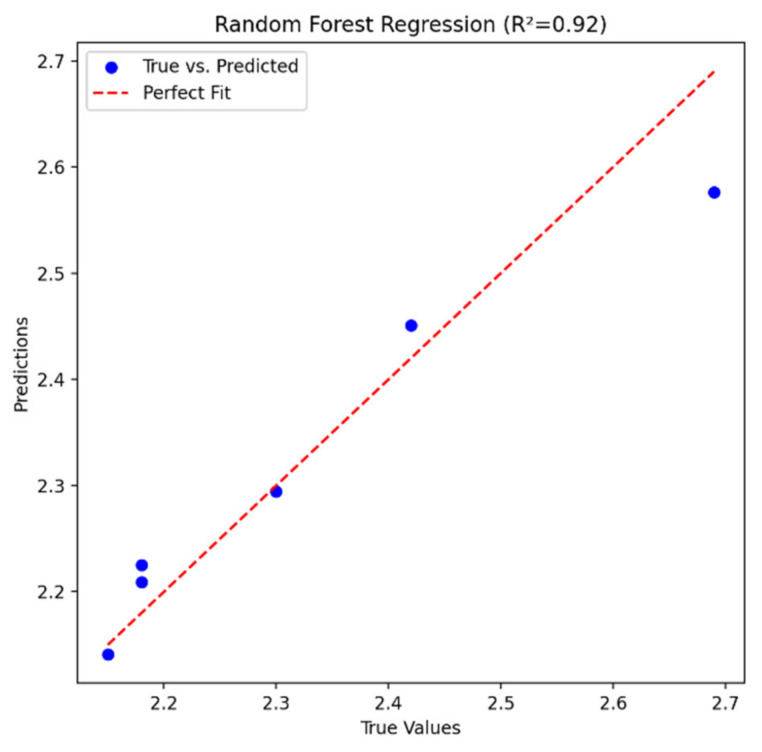
Chlorophyll-a concentration inversion model diagram (salinity factor).

**Figure 13 sensors-24-04181-f013:**
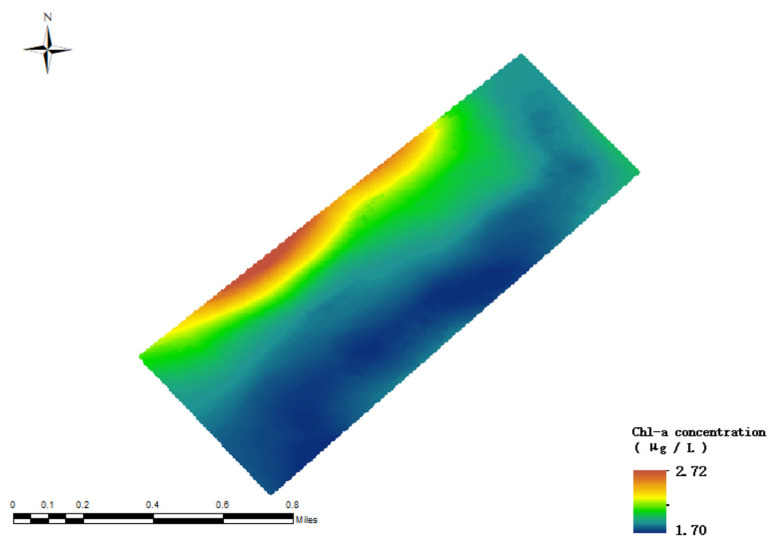
Random forest model inversion results (with the addition of salinity factor).

**Figure 14 sensors-24-04181-f014:**
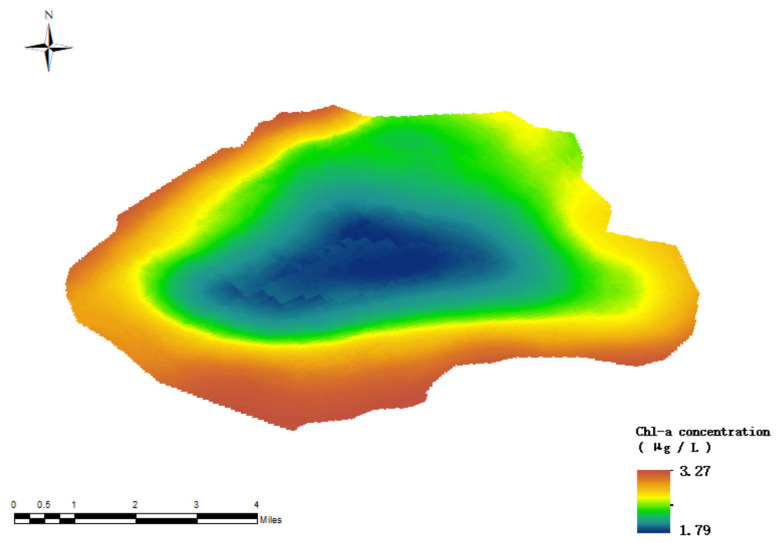
Retrieval of chlorophyll-a concentration in Daihai waters.

**Table 1 sensors-24-04181-t001:** The main parameters of Zhuhai-1 hyperspectral satellite.

Satellite Parameter	Numerical Value
spatial resolution	10 m
width	150 km
quality	67 kg
signal-to-noise ratio	better than 300
operating orbit	98 °o
number of bands	32
spectral resolution	2.5 nm
imaging range	150 km × 2500 km
data efficiency	300 Mbps
wavelength	400 nm–1000 nm
standardization mode	support on-orbit calibration
on-orbit life	more than 5 years

**Table 2 sensors-24-04181-t002:** Introduction of Zhuhai-1 hyperspectral image bands.

Waveband	Center Wavelength	Waveband	Center Wavelength	Waveband	Center Wavelength	Waveband	Center Wavelength
B1	466 nm	B9	596 nm	B17	716 nm	B25	836 nm
B2	480 nm	B10	610 nm	B18	730 nm	B26	850 nm
B3	500 nm	B11	626 nm	B19	746 nm	B27	866 nm
B4	520 nm	B12	640 nm	B20	760 nm	B28	880 nm
B5	536 nm	B13	656 nm	B21	776 nm	B29	896 nm
B6	550 nm	B14	670 nm	B22	790 nm	B30	910 nm
B7	566 nm	B15	686 nm	B23	806 nm	B31	926 nm
B8	580 nm	B16	700 nm	B24	820 nm	B32	940 nm

**Table 3 sensors-24-04181-t003:** Used band information.

Salinity	Wavebands Used
Salinity factor not added	B1, B4, B9, B13, B21, B24, B27, B29
Salinity factor added	B4, B12, B13, B21, B24, B27, B29, B30

**Table 4 sensors-24-04181-t004:** Comparison of inversion model results.

Inversion Model	Salinity	R^2^	RMSE
Random forest model	Salinity factor not added	0.64	0.056
	Salinity factor added	0.92	0.047

**Table 5 sensors-24-04181-t005:** Five-fold cross-validation results of model parameter optimization.

Inversion Model	Salinity	Average R^2^
Random forest model	Salinity factor not added	0.61
	Salinity factor added	0.89

## Data Availability

The data presented in this study are available on request from the corresponding author. The data are not publicly available because they were provided by a government department, and the government department requires confidentiality.

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
