# Peer review of "Inversion Method for Chlorophyll-a Concentration in High-Salinity Water Based on Hyperspectral Remote Sensing Data"

_sensors, 2024, doi:10.3390/s24134181_

Round 1

Reviewer 1 Report (New Reviewer)

Comments and Suggestions for Authors

Comments 1: The principal issue with this paper is the use of the English language. Despite the numerous corrections that have been made, the paper still contains a number of significant flaws.

Comments 2: The title of the second section should be the “Study area and data . This should then be divided into two sections, the first of which should describe the area and the second of which should describe the data.

Comments 3: Figure 2 illustrates the technical route, which is a component of the method. It should be included in the methodological section and should be consistent with the corresponding chapter.

Comments 4: The acquisition of ground data and remote sensing data should be introduced in two distinct sections of a chapter and integrated into a unified whole.

Comments 5: Figure 13 should be included in the Results section and not in the Conclusion section, which should only contain a textual description.

Comments 6: The font size of the latitude and longitude in Figure 3 is relatively small, which has the effect of blurring the image. It is therefore necessary to adjust the font size. Some other figures have similar problems.

Comments 7: Figures 4 and 5 require correction of the orientation of the vertical axis text, which is currently incorrect.

Comments 8: The illustrations in Figure 5 are predominantly qualitative in nature; therefore, it would be beneficial to include some quantitative indicators, such as fitted lines, to enhance the visual representation.

Comments on the Quality of English Language

The quality of English needs improving.

Author Response

Reviewer 2 Report (New Reviewer)

Comments and Suggestions for Authors

The objective of the manuscript is to apply remote sensing technology in high salinity water environments, such as Lake Daihai. A random forest model was built taking salinity into account, so the accuracy of the model improves after adding salinity data to the model. The relationship between salinity and chlorophyll a is well known, so if you want to build a model, it is necessary to introduce salinity. In general, I did not find this study to make a valuable contribution to the literature or to science.

 General comments

In the Introduction section, it is necessary to rewrite the last paragraph (lines 164-181), specifying the objectives or hypotheses of the work.

Conclusions section: rewrite this section indicating the most important results and findings of the work

In the figures, check the direction of the title of the vertical axes

In Figure 4B, review reflectivity values. From 0 to 100%.

In Figure 7, reflectivity as a function of wavelength and not wavenumber.

Author Response

Please refer to the attached document

Reviewer 3 Report (New Reviewer)

Comments and Suggestions for Authors

This manuscript proposes a method for inverting chlorophyll a concentration in Daihai Lake through hyperspectral image, considering the effect of salinity in high-salinity water. The authors demonstrate that adding salinity to the features used by random forest (RF) algorithm can improve the performance of tree models in terms of regression. However, there are problems in the writing, methodology and experiments, while the description of this method needs more details.

1. In Abstract, line 19-20 “the traditional inversion model of chlorophyll a concentration” Could you please describe what does it represent? Is it the method using hyperspectral data? The current description tends the readers to confuse with the above “traditional monitoring methods”.

2. Line 27-28 “The results show ...” and line 28-30 “The R2 value reaches 0.9, indicating that ...” describe virtually the same content, and it is suggested to merge them. In addition, line 29 “better explain the variation of the target variable” and line 30 “further improve the performance of the model” also describe the same content.

3. Introduction lacks logic, and it is suggested to be rewritten. For instance, the authors could describe the association between chlorophyll a and water quality evaluation and then describe the association between chlorophyll a and remote sensing technology. In the current manuscript, they are mixed together, which may confuse the readers. In addition, in Introduction, the description of inversion studies of chlorophyll a concentration requires a clear categorization and a description of representative methods involved, instead of listing them simply without logic. Line 63 “In recent years, many domestic and foreign scholars ... ” Here the description may be unsuitable.

4. In 1 Overview of the study area and data sources, the division of the training and test sets requires to be described in more details. Could you please describe how much sampling points out of the 30 points are used for training and they are selected randomly or manually?

5. In 2 Data processing, line 369-370 “Therefore, it is necessary to screen out the band set that contains the original band information ...” Could you please describe what does it mean? Is it to select a subset of the 32 original bands?

6. In 2 Data processing, lines 375-378 could you please describe whether the salinity is calculated by the conductivity method? I did not find the description in the manuscript.

7. In 3 Random forest model, line 401 “Random forest performs well in dealing with high-dimensional data, ... ” Based on the number of samples and feature dimensions in the data used in this manuscript, it is difficult to describe it as high-dimensional data. Thus, it is suggested to add an experiment that compares the performance between RF and various learning algorithms in terms of regression on this hyperspectral data.

8. In 4.2 Characteristic band analysis, lines 451-453 “In this process, the six bands of B4, B13, B21, B24, B27 and B29 are effectively utilized by the model in both cases.” On the basis of the description in the manuscript, the authors select 8 principal components of the original spectral bands as features to train the model. In addition, could you please describe whether the 32 spectral bands are also used as features to train the model?

9. In 4.5 Cross validation, the K-fold cross validation should be a division for the training set, and this procedure should not involve the test set, especially in the case that there is only one dataset in this manuscript. Please consider this in the experiments and describe it in detail in the manuscript.

10. 4.6 Analysis of effect is a simple repetition of the previous content, and it is suggested to be removed.

11. In 5 Conclusion, line 646-653 it is suggested to remove this part since it describes content that is not relevant to the proposed method.

Author Response

Please refer to the attached document

Reviewer 4 Report (New Reviewer)

Comments and Suggestions for Authors

This study presents a model for inverting chlorophyll a concentration in high salinity waters using hyperspectral remote sensing data from Daihai Lake, Inner Mongolia. A random forest model was developed, demonstrating significant accuracy improvement when salinity was included as a factor.

1. The methodology section lacks detailed explanation on the preprocessing steps and parameter settings used in the random forest model.

2. Limited discussion on the potential biases introduced by the chosen remote sensing data or the random forest algorithm.

3. The literature review, while extensive, misses discussing studies that might have dealt with similar high salinity environments.

4. The study relies solely on Zhuhai-1 satellite data, limiting the generalizability of the findings. A comparison with other remote sensing data or methods is necessary to validate the model's broader applicability.

5. Incomplete comparison with other machine learning models or traditional methods in terms of performance metrics.

6. The paper suffers from coherence issues, especially in transitions between sections. Additionally, figures and tables are not referenced clearly within the text, and there are several grammatical errors and awkward phrasings that detract from the overall clarity.

7. Some important references are missing such as doi: 10.1109/TGRS.2024.3366536,  doi: 10.1109/JSTARS.2024.3350044, doi: 10.1109/TGRS.2023.3318001,doi.org/10.3390/jmse12020357 and doi: 10.1109/TGRS.2023.3348653.

Round 2

Reviewer 2 Report (New Reviewer)

Comments and Suggestions for Authors

In figure 8, the horizontal axis as a function of wavelength

Author Response

Reviewer 3 Report (New Reviewer)

Comments and Suggestions for Authors

The authors have addressed all the issues that I am concerned about.

Author Response

Dear Reviewer,

         I sincerely appreciate the valuable opinions and suggestions you have provided during the review of my manuscript. These insightful insights have not only greatly improved the quality of my paper but also provided me with invaluable academic guidance. I deeply admire your diligent and thorough work ethic as well as your profound academic proficiency.

         I have carefully studied each of your suggestions and attempted to make corresponding revisions and improvements to the paper. These revisions have not only enhanced the logic and rigor of the paper but also made it more compliant with academic norms and standards. I believe that under your careful guidance, my paper has made significant progress.

         Once again, I am grateful for your concern and support for my paper. I am well aware that without your help and guidance, my paper would not have reached its current level.

Thank you.

Wang Nan

Reviewer 4 Report (New Reviewer)

Comments and Suggestions for Authors

The author didn't response to comment 7. 

Author Response

Dear Reviewer,

         I have responded to your seventh suggestion as follows.

         I have downloaded and carefully read all the important references you listed, including "Deep Parameterized Neural Networks for Hyperspectral Image Denoising", "Research on the Drift Prediction of Marine Floating Debris: A Case Study of the South China Sea Maritime Drift Experiment", "Mask Guided Local-Global Attentive Network for Change Detection in Remote Sensing Images", "Material-Guided Multiview Fusion Network for Hyperspectral Object Tracking," and "Spatial–Temporal Siamese Convolutional Neural Network for Subsurface Temperature Reconstruction."

        After thorough research and analysis, I have found that the topics and research directions discussed in these references are not directly related to the content of my manuscript. While these papers possess significant academic value in their respective fields, I was unable to incorporate them into my work due to the specific focus and scope of my paper. I deeply regret not being able to cite these excellent references and would like to express my sincere apology.

       Once again, I appreciate your valuable suggestions, which have been immensely helpful in the preparation of my manuscript. I will continue to work hard to refine my paper and ensure its accuracy and rigor.

This manuscript is a resubmission of an earlier submission. The following is a list of the peer review reports and author responses from that submission.

Round 1

Reviewer 1 Report

Comments and Suggestions for Authors

The paper has low quality, especially for the introduction. The innovation is not enough and the contribution is missing. The presentation of the manuscript is also poor. In summary, the manuscript is difficult to review and process.

Comments on the Quality of English Language

The English language can be understood, but too many errors can be found. The quality needs to improve.